# Hepatic Steatosis and Diet in Adult Celiac Disease: A Cross-Sectional Study

**DOI:** 10.3390/nu17223577

**Published:** 2025-11-15

**Authors:** Míra Zsófia Peresztegi, Zsolt Szakács, Nelli Farkas, Gábor Szekeres, Nándor Faluhelyi, Krisztina Hagymási, Gyula Pásztor, Zsófia Vereczkei, Petra Fülöp, Szilvia Lada, Sarolta Dakó, Eszter Dakó, Judit Bajor

**Affiliations:** 1Institute for Translational Medicine, Medical School, University of Pécs, 7624 Pécs, Hungary; peresztegi.mira@pte.hu (M.Z.P.); nelli.farkas@aok.pte.hu (N.F.) vereczkei47@gmail.com (Z.V.); 2First Department of Medicine, Medical School, University of Pécs, 7624 Pécs, Hungary; szakacs.zsolt@pte.hu (Z.S.); fulop.petra@pte.hu (P.F.); 3Institute of Bioanalysis, Medical School, University of Pécs, 7624 Pécs, Hungary; 4Department of Medical Imaging, Medical School, University of Pécs, 7624 Pécs, Hungary; gabor.szekeres@pte.hu (G.S.); nandor.faluhelyi@pte.hu (N.F.); 5Department of Surgery, Transplantation and Gastroenterology, Faculty of Medicine, Semmelweis University, 1082 Budapest, Hungary; hagymasi.krisztina@med.semmelweis-univ.hu (K.H.); dako.sarolta@gmail.com (S.D.); dako.eszter@phd.semmelweis.hu (E.D.); 6Department of Radiology, Albert Szent-Györgyi Health Centre, University of Szeged, 6720 Szeged, Hungary; pasztor.gyula@szte.hu; 7Department of Sport Nutrition and Hydration, Institute of Nutritional Science and Dietetics, Faculty of Health Sciences, University of Pécs, 7621 Pécs, Hungary; 8Directorate of Nursing Management and Professional Education, Albert Szent-Györgyi Health Centre, University of Szeged, 6725 Szeged, Hungary; lada.szilvia@med.u-szeged.hu

**Keywords:** celiac disease, hepatic steatosis, gluten-free diet, Mediterranean diet

## Abstract

**Background**: Celiac disease (CD) is a chronic immune-mediated enteropathy that is treated exclusively with a lifelong gluten-free diet (GFD). Hepatic involvement, including hepatic steatosis (HS), is common in both newly diagnosed and long-term GFD-treated CD patients. Limited data exist regarding HS prevalence and risk factors in CD, and the effects of dietary patterns, including GFD and the Mediterranean diet (MD), remain unclear. **Objective**: This study investigated the prevalence and severity of HS in newly diagnosed, pre-GFD and GFD-treated CD patients compared to non-celiac control subjects, while assessing the influence of dietary adherence. **Methods**: In a nested cross-sectional study within the ARCTIC trial (NCT05530070), 290 Hungarian adults were enrolled (60 pre-GFD CD, 156 CD on GFD, and 74 control subjects). HS was assessed by ultrasonography, and dietary adherence was evaluated using the Standardized Dietitian Evaluation and Mediterranean Diet Score (MDS). Binary regression models were applied to identify predictors of HS. **Results**: HS was diagnosed in 34% of participants, most frequently in pre-GFD CD patients. BMI was the strongest predictor of HS both overall and within the CD cohort (OR = 1.27; 95% CI: 1.16, 1.42; *p* < 0.001). Neither adherence to GFD nor overall MD adherence significantly influenced the prevalence of HS. Severity of HS correlated with higher BMI, older age, and diabetes prevalence, while individual MD components, including olive oil consumption, were associated with milder HS. **Conclusions**: HS is more prevalent in CD patients, particularly pre-GFD patients, and is strongly associated with BMI. While overall dietary patterns did not significantly impact HS, certain diet components may modulate severity.

## 1. Introduction

Celiac disease (CD) is a chronic immune-mediated enteropathy, for which the only currently known therapy is a strict, lifelong gluten-free diet (GFD). The more well-known symptoms of CD are gastrointestinal manifestations, but extraintestinal manifestations are also common, including hepatic involvement, such as elevated transaminases or hepatic steatosis (HS) [1,2,3].

HS is the abnormal accumulation of fat within hepatocytes. Several potential triggering factors are known, including alcohol consumption; obesity; diabetes mellitus; and metabolic, toxic, and infectious causes [4]. Standard diagnostic methods include imaging techniques such as ultrasound (US), MRI, and CT; however, the gold standard for diagnosis is liver biopsy [5,6]. If left untreated, HS may progress to steatohepatitis, fibrosis, cirrhosis, and eventually hepatocellular carcinoma.

The published data suggest that the prevalence of HS is higher among individuals with CD. A recently conducted meta-analysis found that in adult CD patients, the prevalence of HS is 18.2% before initiation of a GFD and rises to 28.2% after starting the diet [7]. Scientific findings indicate that HS in CD patients is a multifactorial phenomenon [8]. From a pathophysiological perspective, increased intestinal permeability, immune activation (including TLR and inflammasome signalling), altered bile acid metabolism (particularly through FGF19 dysregulation), genetic predisposition, and gut microbiome dysbiosis all contribute to the development of HS [9,10].

Some studies suggest that initiation of a GFD increases the prevalence of HS among CD patients, even though GFDs lead to clinical and histological remission of CD. In general, a GFD is characterized by high caloric and fat contents but low fibre intake, thereby increasing the risk of metabolic syndrome and causing a disproportionate increase in body weight due to a substantial rise in body fat mass [11]. It may result in unfavourable body composition changes, increase the risk of insulin resistance, and obesity is becoming increasingly common among celiac patients on a GFD [12,13]. The same meta-analysis mentioned above concluded that the prevalence of metabolic syndrome is only 4.3% among pre-GFD CD patients, whereas it rises to 21.3% among those adhering to a GFD [7].

Preventing HS involves important lifestyle and dietary factors, with the Mediterranean diet (MD) being the most widely recommended option due to its benefits in preventing cardiovascular and metabolic diseases [14,15]. High-calorie, high-fat Western-style diets—such as GFDs—contribute to the development of HS. Adopting an MD can decrease liver fat even in the absence of weight loss and is regarded as the most recommended dietary approach for HS [16].

To mitigate the unfavourable effects of GFDs, the MD may provide a solution [15,17]. The MD emphasizes high consumption of fish, olive oil, fruits, vegetables, whole grains, legumes, and nuts. This diet is characterized by a lower carbohydrate intake, accompanied by a higher consumption of monounsaturated and omega-3 fatty acids [14,16].

Combining a GFD with the MD could help prevent obesity, metabolic syndrome, and cardiovascular diseases; for this purpose, dietary counselling is essential in managing CD patients. Our research group has previously investigated adherence to the MD in CD patients and found low adherence, with particularly insufficient consumption of whole grains [18].

Currently, limited data are available on how CD precisely influences the development of HS, and there is no clear evidence as to whether a GFD increases or decreases the risk of HS. Understanding the prevalence, mechanisms, diagnostic challenges, and management strategies for HS in CD is essential for optimizing patient outcomes.

The objectives of this study were (1) to investigate the prevalence of HS in celiac patients on a GFD and newly diagnosed celiac patients (pre-GFD) compared to non-celiac control subjects, and (2) to identify predictors of HS, with a special focus on the associations with dietary factors, including adherence to Mediterranean and gluten-free diets.

## 2. Materials and Methods

### 2.1. Study Design

In this nested cross-sectional study of an ongoing prospective, multicentre study (ARCTIC, NCT05530070, three tertiary care centres in Hungary—the Clinical Centre of the University of Pécs, the Department of Surgery, Transplantation and Gastroenterology at Semmelweis University, and the Albert Szent-Györgyi Clinical Centre—enrolled CD patients over the age of 18. The Scientific and Research Ethics Committee of the Hungarian Medical Research Council approved this study (27521-5/2022/EÜIG).

### 2.2. Patient Enrolment

Participants were enrolled in three groups (newly diagnosed, pre-GFD CD patients, CD patients on a GFD for a minimum of 1 year, non-celiac control subjects) between November 2022 and June 2025.

According to current guidelines, the diagnosis of CD in adults requires both positive serological and histological findings [19], which we verified in all enrolled patients. For patients diagnosed during childhood, the diagnosis was verified in accordance with the ESPGHAN guidelines [20].

Healthcare workers and medical students meeting the inclusion criteria were recruited for the control group.

The exclusion criteria for all participants included any acute illness or acute exacerbation of existing chronic conditions, advanced chronic diseases (including heart failure, renal failure, and liver failure), systemic autoimmune diseases, malignant tumours, pregnancy, breastfeeding, and refractory CD. Individuals who declined to sign the informed consent form were also excluded. In the control group, positivity for CD-specific autoantibodies detected during enrolment was considered an exclusion criterion.

More detailed methodological information on the ARCTIC trial has been published elsewhere [21,22].

In this study, data about age, sex, body mass index (BMI), BMI group, presence of diabetes, HS, GFD adherence, and MD adherence were assessed.

### 2.3. Assessment of Hepatic Steatosis

Ultrasonography represents the most suitable screening modality for HS given its short examination time, high sensitivity (82–94%), and non-invasive nature [23].

Ultrasound scans were performed using Samsung RS85 (Samsung Electronics, Suwon, Republic of Korea), GE LOGIQS7 (General Electric, Boston, MA, USA) and E10 systems with convex transducers. The patients fasted for at least 6 h prior to imaging. The examinations were performed in the supine position, with the arms elevated to widen the intercostal acoustic window. Standard intercostal and subcostal windows were used to visualize the right liver lobe and kidney, and also the spleen and the left kidney. Multiple images were acquired to optimize the assessment of echogenicity, vessel visualization, and attenuation.

We used the Hamaguchi ultrasound scoring system [23], with scores ranging from 0 to 6, and the diffuse HS grading system [24], with scores ranging from 0 to 3, to semi-quantitatively assess HS. HS was considered present if either score was greater than zero.

#### 2.3.1. Hamaguchi Score

Liver brightness and hepatorenal echo contrast: Scored 0–3 depending on the intensity of liver parenchymal echogenicity relative to the renal cortex and degree of brightness.

Deep attenuation: Scored 0–2 according to the degree of posterior beam attenuation and the visibility (or lack thereof) of the diaphragm.

Vessel blurring: Scored 0 or 1 depending on whether intrahepatic vessels show blurred boundaries and narrowing of the lumen.

See Appendix A for examples of scoring of abdominal US images.

#### 2.3.2. Diffuse Hepatic Steatosis Grading (US Grade)

Grade 0 (none): Normal liver echotexture, no obvious increase in echogenicity.

Grade 1 (mild): Slight diffuse increase in hepatic echogenicity; intrahepatic vessels and diaphragm remain clearly visible.

Grade 2 (moderate): Moderate increase in echogenicity; some impairment in visualizing vessels or diaphragm.

Grade 3 (severe): Marked increase in echogenicity with poor or absent visualization of vessels, diaphragm, or posterior liver segments.

### 2.4. Dietary Evaluation

Adherence to a GFD was evaluated by a trained dietitian using the Standardized Dietitian Evaluation (SDE) method [25].

Compliance with the MD was assessed using the Mediterranean Diet Score (MDS), a validated 14-item questionnaire commonly employed as an indicator of healthy dietary patterns [26] (Appendix A). The score ranges from 0 to 14 depending on the consumption of key MD components such as vegetables, fish, and olive oil. Additionally, the individual components of the MDS were analysed to evaluate the adequacy of intake in each group: a score of 1 denoted adequate intake, while values of 0 or 0.5 indicated inadequate intake.

### 2.5. Statistical Analysis

Descriptive statistical tools were used to summarize the demographic and clinical characteristics of the study population. Continuous variables were expressed as means ± standard deviations (SDs), while categorical variables are presented as frequencies and percentages. Group comparisons were performed using the Chi-square test or Fisher’s exact test for categorical variables, and the Kruskal–Wallis test for continuous variables.

To identify the risk factors for HS, both machine learning analysis (random forest classifier) and binary logistic regression were performed. The machine learning analysis was used to explore variable importance, and the results were integrated into the regression models. Three separate models were created: Model 1 included age, sex, diabetes, BMI, SDE score, and the total MDS as variables; Model 2 additionally incorporated the individual components of the MDS; and in Model 3, the total MDS was excluded. The same models were applied to assess the predictors of HS specifically within the celiac population. In all cases, odds ratios (ORs) and 95% confidence intervals (CIs) were calculated.

All statistical analyses were performed using R software (version 4.2.2, R Core Team, R Foundation for Statistical Computing, Vienna, Austria. URL: https://www.R-project.org/, accessed on 5 September 2025). A *p*-value < 0.05 was considered statistically significant.

## 3. Results

A total of 290 individuals were enrolled in the study: 60 pre-GFD CD patients, 156 CD patients on a GFD, and 74 control subjects (Table 1). The mean age was 35.3 years (±12.5), with a range of 18 to 76 years. The mean age differed significantly across the three groups: CD patients on a GFD had the highest median age, while the control group had the lowest (Table 1). Of the patients, 237 were female (82%). Diabetes mellitus was present in nine cases (3.1%). The overall mean BMI was 23.7 (±5.3), with values of 23.8 (±6.3) in the pre-GFD patients, 23.7 (±5.5) in the CD patients on a GFD, and 23.6 (±4.0) in the control group (*p* = 0.665 for difference across the groups). Regarding BMI category, 30 (10%) individuals were underweight, 165 (57%) had a normal weight, 62 (21%) were overweight, and 33 (11%) were obese.

HS was diagnosed in 99 patients (34%), including 18 with moderate and 6 with severe disease. When comparing those with and without HS, no significant differences were observed between the groups with respect to sex, diabetes prevalence, BMI, or BMI categories (*p* > 0.05 for all comparisons) (Table 2).

Among the participants with HS, 27/30 (90%) of the newly diagnosed CD patients, 42/54 (78%) of the CD patients on a GFD, and 13/15 (87%) of controls met the criteria for MASLD.

MDSs were significantly higher in the non-celiac control subjects compared with the pre-GFD CD patients (5.5 vs. 6.0; *p* = 0.049). Whole-grain consumption was also higher in the control group relative to both CD patient groups (on a GFD and pre-GFD) (*p* < 0.001 and *p* = 0.008, respectively).

### 3.1. Predictors of Hepatic Steatosis

HS was most common in the pre-GFD CD patients, whereas the control subjects had the lowest prevalence (Table 2). Ultrasound grades were higher in the pre-GFD CD patients compared with the control group (*p* = 0.018). The Hamaguchi Score was lower in the control group than in both patient groups (control group vs. CD on GFD, *p* = 0.037; control group vs. pre-GFD CD, *p* < 0.001).

The participants diagnosed with HS were older (*p* < 0.001) and had higher BMI values (*p* < 0.001) than the participants without HS.

When stratifying by disease duration, the patients with HS had a significantly shorter duration of CD compared to those without HS (median: 5.5 years [IQR: 2–9.75] vs. 8 years [IQR: 3–14], *p* = 0.048).

Olive and rapeseed oil consumption was closer to the optimal level among those without HS (*p* = 0.023). No significant differences were observed with respect to sex, diabetes, MDS components (vegetable intake, legumes and nuts, fruit intake, whole grains, lean meats, fish, fatty meats, dairy products, vegetable oils, avocado, sugar, or wine consumption) (*p* > 0.05 in all cases). Similarly, there were no differences in total MDS or SDE scores (*p* > 0.05 in all cases).

Using machine learning methods, we identified the key parameters associated with the development of HS (Appendix A). Then, based on these results, we constructed three binary logistic regression models. Models 1 and 3 identified BMI (OR = 1.20; 95% CI: 1.13, 1.29; *p* < 0.001) and group allocation (presence of CD and whether patients were on a GFD or pre-GFD; control group vs. pre-GFD CD: OR = 4.72, 95% CI: 2.09, 11.1, *p* < 0.001; control group vs. CD on GFD: OR = 2.11, 95% CI: 1.06, 4.41, *p* = 0.040) as significant predictors, whereas Model 2 additionally identified diabetes mellitus (OR = 5.29; 95% CI: 1.07, 30.6; *p* = 0.045) as a significant positive predictor of HS (Table 3). All three models were further refined using the stepAIC method, yielding similar results.

### *3.2.* Predictors of Hepatic Steatosis Among Celiac Patients

The CD patients with HS had higher BMI values than those without HS (*p* < 0.001), and overweight and obesity were more common in the group with HS (*p* < 0.001). No differences were observed in SDE scores between the celiac patients adhering to a GFD with and without HS (*p* > 0.05) (Table 2).

BMI emerged as the strongest and most significant positive predictor of HS (OR = 1.27; 95% CI: 1.16, 1.42; *p* < 0.001), with each one-unit increase in BMI associated with an approximately 27% higher odds of HS (the BMI range of the subjects in the study: 11.9–47.8) (Appendix A). In Models 2 and 3, whole-grain consumption was significantly associated with HS (Model 2: *p* = 0.031; Model 3: *p* = 0.020), with odds ratios of 5.50 (95% CI: 1.18, 27.5) and 6.27 (95% CI: 1.35, 31.1), respectively, indicating that the celiac patients that were consuming whole grains had over a five-fold increased risk of HS. In contrast, diabetes was not a significant factor in this population (Model 2: *p* = 0.431; Model 3: *p* = 0.407).

### 3.3. Predictors of the Severity of Hepatic Steatosis

Although the severity of HS did not differ significantly across groups (*p* = 0.388), the highest proportion of moderate and severe cases was observed among the CD patients on a GFD (5.8% and 3.2%, respectively) (Table 1).

The patients with severe HS were the oldest (Appendix A). The prevalence of diabetes was higher among the patients with severe HS compared to those without HS or with mild HS (*p* = 0.007 and *p* = 0.042, respectively). Increasing HS severity was associated with higher BMI in all the groups (Appendix A).

High-fat, non-fermented dairy consumption was higher in the patients with mild and severe HS, as well as in the patients without HS, compared with those with moderate HS (mild HS vs. moderate HS *p* = 0.037; severe HS vs. moderate HS *p* = 0.011; without HS vs. moderate HS *p* = 0.010). Olive and rapeseed oil intake was also higher in patients without HS compared with the patients with moderate HS (*p* = 0.044).

SDE scores were lower in the patients with mild HS than in those without HS (*p* = 0.045). No significant differences were observed in the severity of HS with respect to sex, total MDS, or MDS components, including vegetable intake, legumes and nuts, fruit intake, whole grains, lean meats, fish, fatty meats, low-fat and fermented dairy products, vegetable oils, avocado, sugar, or wine consumption (*p* > 0.05 in all cases).

## 4. Discussion

Our results indicate that the prevalence of HS is significantly higher in individuals with CD compared to non-celiac control subjects, with the highest prevalence observed among pre-GFD CD patients. We observed that dietary patterns, including adherence to the MD or a GFD, were not significantly associated with the presence of HS.

In the literature, the prevalence of HS among celiac patients who have not yet started a GFD ranges from 1.7% to 18.2%, whereas in celiac patients adhering to a GFD, it ranges from 11.1% to 29.5% [7,12,27]. In our cohort, the HS prevalence was 50% of the pre-GFD CD patients, 35% of the CD patients on a GFD, and 20% of the control subjects; thus, our findings appear to contradict previous reports. The significant difference was confirmed by both group comparison tests and binary models.

The high prevalence of HS observed in the pre-GFD CD patients may be explained by the observation that in conditions associated with protein malabsorption, such as after jejunoileal bypass surgery [28] or in kwashiorkor [29], a deficiency of lipotropic factors is present, which—when combined with pyridoxine deficiency—may lead to HS. In addition, alterations in the intestinal microflora have been described in malnutrition, and dysbiosis has been shown to contribute to the development of HS [30]. Patients with CD have an abnormal gut microbiota characterized by low levels of Lactobacilli and Bifidobacteria, as well as an increase in Gram-negative bacteria [31,32,33]. It has also been established that a GFD leads to a reduction in the concentration of both potential pathogenic bacteria and Bifidobacteria [31]. Alterations in gut microbiota associated with both CD and a GFD might influence the gut–liver axis, facilitating HS [8,32].

Moreover, the link between small intestinal bacterial overgrowth (SIBO) and CD is observed primarily in pre-GFD CD patients before initiating a GFD [34]. The proinflammatory responses and leaky gut syndrome characteristic of pre-GFD CD may also contribute to the higher prevalence of HS observed in pre-GFD CD patients.

Multiple studies suggest that altered intestinal permeability may contribute to the development of HS [35]. Endotoxins produced by gut bacteria, such as lipopolysaccharides, can translocate to the liver and activate immune responses via Toll-like receptors [36]. Diet-induced improvements in intestinal mucosal inflammation, villous atrophy, and permeability may explain the observed decrease in HS incidence following the initiation of a GFD.

There are other interconnected factors associated with the development of HS in CD patients. Metabolic dysfunction may result from several mechanisms: increased nutrient absorption following mucosal atrophy improvement on a GFD; hyperphagic compensatory status after a malabsorption disorder; and high intake of calories, fats, and sugars [37,38]. Additionally, an Italian prospective interventional clinical study found that a higher relative intake of packaged gluten-free foods was significantly correlated with the presence of HS [39]. The low-grade chronic inflammation present in CD—even persisting during a GFD—also contributes to the metabolic dysfunction in CD [31,40].

Our results confirmed previous findings indicating that a higher BMI is associated with an increased risk of HS [41]. The analysis conducted exclusively within the celiac cohort was consistent with this result, as BMI was the only factor showing a significant association with HS.

In this study, we hypothesized that diet may play a role in steatogenesis in celiac patients; therefore, we examined both GFDs and the MD. In our previous publication, we analysed adherence to the MD among CD patients in detail [18]. In the present cohort, with a larger sample size, we observed a significant difference between pre-GFD CD patients and non-celiac control subjects: according to MD scores, celiac patients demonstrated worse adherence to the MD.

However, we were unable to demonstrate a significant effect of either diet on the development of HS. At the same time, among the individual elements of the MD, the consumption of olive and rapeseed oil showed a significant association with the presence and severity of HS. This finding suggests a potential beneficial effect of olive and rapeseed oil on HS, although this association requires further clarification in future studies.

The finding that whole-grain consumption was associated with a higher prevalence of HS in celiac patients was highly unexpected. This result is in clear contradiction with previous evidence suggesting that whole-grain intake is beneficial in metabolic and metabolic-associated diseases. We were unable to identify an explanation for this observation.

We also examined the impact of GFD adherence on the development and severity of HS among CD patients adhering to the diet for at least one year, but no significant associations were found. Previous studies have shown that a GFD improves the proinflammatory response and reduces intestinal permeability, commonly referred to as “leaky gut” [42]. However, despite being the only available treatment, a GFD—given its suboptimal nutrient composition—may have adverse metabolic consequences, leading to unfavourable changes in body composition. Improved nutrient absorption following the introduction of a GFD promotes weight gain, but fat mass increases to a greater extent than lean mass, which may predispose to HS. Our study also found a higher BMI to be the strongest risk factor for HS; nonetheless, our findings indicate that the prevalence of HS is lower among patients on a GFD compared to pre-GFD CD patients.

Despite the potential drawbacks of a GFD, we maintain that eliminating the proinflammatory response through a GFD should remain the primary therapeutic goal. Our findings suggest that HS may occur more frequently during the earlier stages of the disease. This may be due to malabsorption and low-grade inflammation, which can persist for years after starting a GFD. At the same time, it is essential to address the unfavourable metabolic consequences of GFDs and mitigate the adverse changes in body composition. During the follow-up, it is worth paying attention to prevent undesirable weight gain. For this reason, dietary interventions should play an important role in patient management. The effectiveness and specific impact of such interventions are being investigated in our ongoing ARCTIC study (NCT05530070), the results of which will hopefully provide further insights into this topic.

### Strengths and Limitations

To the best of our knowledge, this is the first study to investigate the associations between HS and both pre-GFD and on-GFD CD in this form, incorporating dietary habits into the analysis. Another strength is the focus on dietary components, which allowed us to identify potential nutritional factors associated with HS within this specific patient population. The high quality of the data and the reliability of our analytical strategy further reinforce the credibility of our findings. Finally, the fact that BMI appeared as a significant predictor—consistent with numerous previous studies—supports the validity of our analytical strategy as an external standard.

Several limitations should be acknowledged. Some analyses were underpowered, and the range of assessed parameters was limited; factors such as physical activity and comorbidities, which may influence hepatic steatosis, were not included. The lack of liver biopsy (which is considered the gold standard for determining HS) is another limitation; however, ultrasonography is a validated and widely accepted non-invasive alternative for assessing HS in both clinical and research settings [43,44].

Additionally, although the method used to assess Mediterranean diet adherence does not capture all aspects of dietary behaviour, it remains the most reliable approach that is currently available. Another limitation is that alcohol intake was assessed using the MDS, which does not allow for the precise quantification of consumption; therefore, we could not fully differentiate alcohol-related from non-alcohol-related hepatic steatosis, in line with the evolving MASLD/MetALD framework that recognizes overlapping aetiologies.

## 5. Conclusions

In conclusion, HS is more prevalent among CD patients compared to non-celiac control participants, particularly those who are newly diagnosed and not yet on a GFD. The dietary adherence—whether to a GFD or the MD—did not prove to be a significant predictor of HS and its severity. Further studies are needed to clarify the associations between CD and HS, as well as between the dietary patterns of CD patients and HS.

## Figures and Tables

**Table 1 nutrients-17-03577-t001:** Characteristics of the study population.

		CD on GFD	Pre-GFD CD	Controls	CD on GFD vs. Pre-GFDCD	CD on GFD vs. Controls	Pre-GFD CD vs. Controls
					*p*-Value		
**Number of subjects** (n)		156	60	74			
**Age at enrolment** mean ± SD		36 (±13)	37 (±13)	32 (±11)	>0.9	**0.012**	**0.020**
**Sex** n (%)	Women	127 (81%)	51 (85%)	59 (80%)	0.7	0.9	0.5
	Men	29 (19%)	9 (15%)	15 (20%)			
**Diabetes** n (%)		6 (3.3%)	2 (3.3%)	1 (1.4%)	>0.9	0.4	0.6
**BMI** (continuous) mean ± SD		23.7 (±5.5)	23.8 (±6.3)	23.6 (±4.0)	0.9	0.4	0.6
**BMI groups**	**Under-weight**	15 (9.6%)	10 (17%)	5 (6.8%)	0.3	0.1	0.2
	**Normal**	90 (58%)	30 (50%)	45 (61%)			
	**Over-weight**	28 (18%)	14 (23%)	20 (27%)			
	**Obese**	23 (15%)	6 (10%)	4 (5.4%)			
**Hepatic steatosis** n (%)		54 (35%)	30 (50%)	15 (20%)	**0.043**	**0.031**	**<0.001**
**US grade**	**0**	107 (69%)	36 (60%)	60 (81%)	0.4	0.2	**0.018**
	**1**	37 (24%)	21 (35%)	12 (16%)			
	**2**	6 (3.9%)	2 (3.3%)	2 (2.7%)			
	**3**	5 (3.2%)	1 (1.7%)	0 (0%)			
**Hamaguchi Score**	**0**	108 (69%)	31 (52%)	64 (86%)	0.057	**0.037**	**<0.001**
	**1**	26 (17%)	20 (33%)	4 (5.4%)			
	**2**	8 (5.1%)	4 (6.7%)	1 (1.4%)			
	**3**	4 (2.6%)	4 (6.7%)	4 (5.4%)			
	**4**	5 (3.2%)	1 (1.7%)	1 (1.4%)			
	**5**	2 (1.3%)	0 (0%)	0 (0%)			
	**6**	3 (1.9%)	0 (0%)	0 (0%)			
**Severity of hepatic steatosis**	**mild**	40 (26%)	25 (42%)	10 (14%)	0.388		
	**moderate**	9 (5.8%)	4 (6.7%)	5 (6.8%)			
	**severe**	5 (3.2%)	1 (1.7%)	0 (0%)			
**MDS** median (IQR)		5.25 (4.50, 6.50)	5.50 (4.25, 6.50)	6.00 (4.50, 7.50)	0.6	0.073	**0.049**

CD: celiac disease, GFD: gluten-free diet, SD: standard deviation, BMI: Body Mass Index, US: ultrasound, MDS: Mediterranean Diet Score, IQR: interquartile range. Statistical test: Kruskal–Wallis rank sum test, Chi-squared test, Fisher’s exact test as appropriate, level of significance: *p* < 0.05. *p*-values indicate a difference across the three groups. Significant results are highlighted with bold numbers.

**Table 2 nutrients-17-03577-t002:** Predictors of hepatic steatosis.

		Participants Without HS	Participants with HS	*p*-Value
**Group** (n)	**Pre-GFD CD patients**	30 (16%)	30 (30%)	**0.001**
	**CD patients on a GFD**	102 (53%)	54 (55%)	
	**Controls**	59 (31%)	15 (15%)	
**Age at enrolment** mean ± SD		33 (±11)	39 (±14)	**<0.001**
**Sex** n (%)	**Women**	157 (82%)	80 (81%)	0.771
	**Men**	34 (18%)	19 (19%)	
**Diabetes** n (%)		3 (1.6%)	6 (6.1%)	0.067
**BMI** (continuous) mean ± SD		22.2 (±3.6)	26.5 (±6.8)	**<0.001**
**BMI groups**	**Underweight**	23 (12%)	7 (7.1%)	**<0.001**
	**Normal**	131 (69%)	34 (34%)	
	**Overweight**	31 (16%)	31 (31%)	
	**Obese**	6 (3.1%)	27 (27%)	
**MDS** mean ± SD		5.79 (±1.77)	5.42 (±1.48)	0.115
**Vegetables** n (%)	0	24 (13%)	16 (16%)	0.316
	0.5	134 (70%)	72 (73%)	
	1	33 (17%)	11 (11%)	
**Legumes and nuts** n (%)	0	86 (45%)	48 (48%)	0.635
	0.5	84 (44%)	38 (38%)	
	1	21 (11%)	13 (13%)	
**Fruits** n (%)	0	36 (19%)	21 (21%)	0.463
	0.5	106 (55%)	59 (60%)	
	1	49 (26%)	19 (19%)	
**Whole grains** n (%)	0	106 (55%)	58 (59%)	0.814
	0.5	65 (34%)	30 (30%)	
	1	20 (10%)	11 (11%)	
**Lean meat** n (%)	0	16 (8.4%)	13 (13%)	0.396
	0.5	109 (57%)	51 (52%)	
	1	66 (35%)	35 (35%)	
**Fish** n (%)	0	143 (75%)	71 (72%)	0.772
	0.5	40 (21%)	24 (24%)	
	1	8 (4.2%)	4 (4.0%)	
**Fatty meat and processed meat** n (%)	0	45 (24%)	24 (24%)	0.848
	0.5	74 (39%)	41 (41%)	
	1	72 (38%)	34 (34%)	
**High-fat, non-fermented dairy products** n (%)	0	36 (19%)	16 (16%)	0.403
	0.5	62 (32%)	40 (40%)	
	1	93 (49%)	43 (43%)	
**Low-fat fermented dairy products** n (%)	0	81 (42%)	38 (38%)	0.260
	0.5	81 (42%)	51 (52%)	
	1	29 (15%)	10 (10%)	
**Vegetable oils** n (%)	0	52 (27%)	28 (28%)	0.697
	0.5	122 (64%)	65 (66%)	
	1	17 (8.9%)	6 (6.1%)	
**Olive and rapeseed oil** n (%)	0	105 (55%)	70 (71%)	**0.023**
	0.5	74 (39%)	27 (27%)	
	1	12 (6.3%)	2 (2.0%)	
**Avocado** n (%)	0	154 (81%)	86 (87%)	0.107
	0.5	36 (19%)	11 (11%)	
	1	1 (0.5%)	2 (2.0%)	
**Sugar** n (%)	0	22 (12%)	12 (12%)	0.953
	0.5	69 (36%)	34 (34%)	
	1	100 (52%)	53 (54%)	
**Wine** n (%)	0	114 (60%)	68 (69%)	0.323
	0.5	42 (22%)	17 (17%)	
	1	35 (18%)	14 (14%)	
**SDE** (Only in CD patients on a GFD)	0	0 (0%)	1 (1.9%)	0.176
	1	51 (50%)	21 (39%)	
	2	39 (38%)	22 (41%)	
	3	12 (12%)	8 (15%)	
	4	0 (0%)	0 (0%)	
	5	0 (0%)	2 (3.8%)	
	6	0 (0%)	0 (0%)	

CD: celiac disease, GFD: gluten-free diet, SD: standard deviation, BMI: Body Mass Index, SDE: Standardized Dietitian Evaluation. Statistical test: Kruskal–Wallis rank sum test, Chi-squared test, Fisher’s exact test as appropriate, level of significance: *p* < 0.05. *p*-values indicate a difference across the groups. Significant results are highlighted with bold numbers.

**Table 3 nutrients-17-03577-t003:** Binary logistic regression model (Model 2) identifying predictors of hepatic steatosis.

Variable		N	Cases	OR	95% CI	*p*-Value
**Groups**	**Controls**	74	15	1.00	-	
	**pre-GFD CD**	60	30	4.57	1.91; 11.4	**<0.001**
	**CD on GFD**	156	54	2.09	0.991; 4.62	0.059
**Age**		290	99	1.02	0.995; 1.04	0.125
**Sex**	**Male**	53	19	1.00	-	
	**Female**	237	80	1.58	0.749; 3.45	0.236
**Diabetes mellitus**	**No**	281	93	1.00	-	
	**Yes**	9	6	5.29	1.07; 30.6	**0.045**
**BMI (continuous)**		290	99	1.20	1.13; 1.29	**<0.001**
**Vegetables**	0	246	88	1.00	-	
	1	44	11	0.536	0.194; 1.37	0.207
**Fruits**	0	222	80	1.00	-	
	1	68	19	0.779	0.361; 1.64	0.516
**Whole grains**	0	259	88	1.00	-	
	1	31	11	2.31	0.819; 6.46	0.108
**Fish**	0	278	95	1.00	-	
	1	12	4	1.67	0.366; 6.88	0.486
**Fatty meat and processed meat**	0	184	65	1.00	-	
	1	106	34	0.901	0.466; 1.73	0.754
**High-fat, non-fermented dairy products**	0	154	56	1.00	-	
	1	136	43	0.909	0.492; 1.68	0.761
**Olive and rapeseed oil**	0	276	97	1.00	-	
	1	14	2	0.345	0.038; 1.89	0.272
**Sugar**	0	137	46	1.00	-	
	1	153	53	0.967	0.512; 1.82	0.918
**Wine**	0	241	85	1.00	-	
	1	49	14	1.20	0.502; 2.79	0.677
**MDS**		290	99	0.901	0.688; 1.17	0.443

OR: Odds Ratio, CI: Confidence Interval, CD: celiac disease, GFD: gluten-free diet, BMI: Body Mass Index, MDS: Mediterranean Diet Score, Cases: number of hepatic steatosis cases. Statistical test: binary logistic regression model, level of significance: *p* < 0.05. Significant results are highlighted with bold numbers.

## Data Availability

The data presented in this study are available within the article and the Appendix A. The raw data are available on request from the corresponding author.

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
