# Peer review of "Nutrients2025, 17(22), 3577;https://doi.org/10.3390/nu17223577"

_nutrients, 2025, doi:10.3390/nu17223577_

Round 1

Reviewer 1 Report

Comments and Suggestions for Authors

The study addresses the effects of a gluten-free diet on the metabolism of celiac patients.

The study is well-designed and rigorous, yet it is difficult to draw conclusions when too many factors, not all of which can be assessed, are involved.

The authors only manage to demonstrate the higher prevalence of HS in celiac patients, both in GFD and pre-GFD, compared to non-celiac subjects.

The topic remains interesting and worthy of publication, representing another piece of the puzzle in our understanding of celiac disease.

I have highlighted the lack of the underlined numbers in references 8, 9, and 18:

No. 8 Nutrients 2024, 17(1):85

No. 9 Nutrients 2018, 10(9):1239

No. 18 Nutrients 2025, 17(5):788

Author Response

Comments 1: 

The study addresses the effects of a gluten-free diet on the metabolism of celiac patients.

The study is well-designed and rigorous, yet it is difficult to draw conclusions when too many factors, not all of which can be assessed, are involved.

The authors only manage to demonstrate the higher prevalence of HS in celiac patients, both in GFD and pre-GFD, compared to non-celiac subjects.

The topic remains interesting and worthy of publication, representing another piece of the puzzle in our understanding of celiac disease.

I have highlighted the lack of the underlined numbers in references 8, 9, and 18:

No. 8 Nutrients 2024, 17(1):85

No. 9 Nutrients 2018, 10(9):1239

No. 18 Nutrients 2025, 17(5):788

Response 1: 

We thank the reviewer for the positive evaluation and thoughtful feedback. We agree that multiple factors influence the development of hepatic steatosis in celiac disease, and our findings contribute one step toward understanding this complex relationship. We have also carefully revised the reference list and corrected the missing underlined numbers in references 8, 10, and 18 as indicated.

Reviewer 2 Report

Comments and Suggestions for Authors

This is cross sectional study on the steatosis and diet in Celiac Disease. Although the problem itself is interesting and important, there are some limitstions:

  • tables are illegible
  • as far as I understood BMI was compared only in CD group.  I couldn’t find BMI assessments in control group. If BMI in control group is not included how can we know for sure that steatosis is not higher in obese patients regsrdless of diagnosis?
  • Authors themselves agree that liver biopsy is a gold diagnostic standard in steatosis, it was not performed
  • study was conducted among Adults only which should be included ten the title 
  • no information about time duration of CD was given, it May influence the results
  • in children CD May be diagnosed without biopsy under certain conditions so patients enrilled into study have probably never had microscopic confirmation while Abduls always have. I Think it should be better to exclude patients diagnosed in childhood 
  • in my opinion information about Marsh would be interestion
Comments on the Quality of English Language

English is correct

Author Response

Comments 1: tables are illegible

Response 1: 

We thank the reviewer for their comment. The tables were formatted according to the publisher’s manuscript template and formatting standards. Adjusting their size or layout further would result in loss of data.

Comments 2: as far as I understood BMI was compared only in CD group.  I couldn’t find BMI assessments in control group. If BMI in control group is not included how can we know for sure that steatosis is not higher in obese patients regsrdless of diagnosis?

Response 2: 

We thank the reviewer for this valuable observation. BMI data were collected and analyzed for all study participants, including the control group, as presented in row 7 of Table 1 of the manuscript. The mean BMI and BMI categories are shown for each group (CD on GFD: 23.7 (±5.5), pre-GFD CD: 23.8 (±6.3), and controls: 23.6 (±4.0)), and no significant differences were observed across groups (p>0.05). Additionally, BMI was included as a covariate in our binary logistic regression models assessing predictors of hepatic steatosis across all participants. This approach allowed us to account for the potential confounding effect of BMI on hepatic steatosis, independent of celiac disease status.

Comments 3: Authors themselves agree that liver biopsy is a gold diagnostic standard in steatosis, it was not performed

Response 3: 

We thank the reviewer for this observation. We have added a sentence at the end of the Limitations section, as well as a paragraph in the Discussion to clarify this issue. We agree that liver biopsy remains the gold standard for diagnosing hepatic steatosis and fibrosis. However, it has important limitations, including invasiveness, sampling error due to the structural heterogeneity of diffuse liver diseases, inter-reader variability, and potential complications. These factors, as well as ethical considerations, make it unsuitable for screening or for studies requiring repeated assessments.

In our study, we employed ultrasonographic evaluation using the Hamaguchi scoring system, which offers a non-invasive, safe, and widely available alternative. The Hamaguchi score enhances the sensitivity and specificity for identifying metabolic dysfunction–associated steatotic liver disease (MASLD) (>10% steatosis) to 97% and 100%, respectively, compared with liver biopsy. Moreover, this method reduces operator dependency and inter-observer variation, demonstrating accuracy comparable to liver computed tomography.

Ultrasonography is also widely accepted in previous studies as a valid method for the assessment of hepatic steatosis (e.g., DOI: 10.1002/dmrr.3787; DOI: 10.3389/fmed.2024.1425145), and it was the imaging modality predefined in our published study protocol (NCT05530070). While advanced modalities such as MRI-PDFF or FibroScan-CAP can also provide accurate quantification, their high cost and limited availability restrict their use in routine clinical and research settings. Therefore, ultrasonography remains the most feasible, validated, and ethically appropriate method for non-invasive assessment of hepatic steatosis in our study population.

Comments 4: study was conducted among Adults only which should be included ten the title 

Response 4:

We thank the reviewer for this helpful suggestion. We have added the word “adult” to the title to clarify the study population.

Comments 5: no information about time duration of CD was given, it May influence the results

Response 5: 

We thank the reviewer for this important observation. The median disease duration was calculated for both steatotic and non-steatotic CD groups. Patients with HS had a significantly shorter disease duration compared to those without HS (median 5.5 years [IQR: 2–9.75] vs. 8 years [IQR: 3–14], p = 0.048). Notably, in our cohort, the highest prevalence of HS was observed among newly diagnosed CD patients.

Comments 6: in children CD May be diagnosed without biopsy under certain conditions so patients enrilled into study have probably never had microscopic confirmation while Abduls always have. I Think it should be better to exclude patients diagnosed in childhood 

Response 6: 

We thank the reviewer for this thoughtful comment. The diagnosis of celiac disease in all included patients was verified according to the prevailing guidelines at the time of diagnosis. For patients diagnosed before 2012, an intestinal biopsy was performed in all cases. After 2012, the diagnosis could also be established based on serological criteria, in accordance with the updated ESPGHAN guidelines, as serological markers have been shown in several studies to have a high predictive value for villous atrophy.

Giersiepen, K., et al., Accuracy of diagnostic antibody tests for coeliac disease in children: summary of an evidence report. J Pediatr Gastroenterol Nutr, 2012. 54(2): p. 229-41.

Sheppard, A.L., et al., Systematic review with meta-analysis: the accuracy of serological tests to support the diagnosis of coeliac disease. Aliment Pharmacol Ther, 2022. 55(5): p. 514-527.

Comments 7: in my opinion information about Marsh would be interestion

Response 7:

We thank the reviewer for this valuable suggestion. Unfortunately, the limited time available for the revision did not allow us to collect complete Marsh classification data for all patients. However, we fully agree that this information would provide additional insight, and we plan to include it in our future studies.

Reviewer 3 Report

Comments and Suggestions for Authors

This is a cross-sectional study evaluating hepatic steatosis (HS) by ultrasound in adults with celiac disease (newly diagnosed and on GFD) vs. controls, in order to explore clinical and dietary predictors. The study suggests that body mass index (BMI) was the strongest predictor for HS and that adherence to a GFD or Mediterranean diet (MD) did not significantly affect HS prevalence, though some MD components (e.g., olive oil) were associated with milder disease.

Comments

1. The authors should take into consideration the recent shift in the nomenclature from NAFLD to MASLD/MASH. The authors reported HS prevalence but they did not take into consideration the MASLD criteria (e.g., explicit capture of BP, fasting glucose, TG/HDL, or their treatments), thereby limiting comparability to current guidelines and to prior epidemiology. I would recommend that the authors should consider re-analyzing the prevalence using MASLD definitions and reporting overlap etiologies/exclusions.

2. Based on the previously published literature, the highest incident NAFLD/MASLD risk is reported within the first year after CD diagnosis, persisting long-term, and implicating weight gain and metabolic syndrome after starting GFD. The authors should stratify HS by the time since CD diagnosis and by GFD duration so that they would directly test these signals and better understand the higher HS proportion in pre-GFD patients. 

3. In previously published studies, multiple MASLD pathways were reported that pertained to CD—dietary excess sugar/fat, low fibre, insulin resistance, PPI exposure, physical inactivity, alcohol, and dysbiosis. The authors acknowledged missing key covariates (e.g., physical activity, comorbidities) and found that diabetes was significant in one model; however, should also assess HOMA-IR, PPI use, alcohol intake, or socioeconomic factors, and consider performing a sensitivity analysis including them.

4. We already know that GFDs often have higher sugar/fat and lower fibre/micronutrients, with processed gluten-free products as a distinct exposure. A brief MDS and a general “standardized dietitian evaluation” may miss these gradients. I recommend that the authors: a. quantify total energy, macronutrients (especially free sugars, saturated fat), fibre, and the proportion of ultra-processed gluten-free foods; b. report oats/pseudocereals use separately; and c. consider an a-priori “healthy GFD” score aligned with the current recommendations.

5. The authors' suggestion of non-significant trends (e.g., higher odds with whole grains; favorable pattern with olive/rapeseed oil) seems under-powered. Given prior recommendations encouraging naturally gluten-free whole grains and unsaturated fats for MASLD prevention, I think that the authors should avoid over-stating these exploratory signals and rather present them as hypothesis-generating and adjust for multiple testing.

6. Previous studies recommended serial serology (tTG/EMA) and, where possible, gluten immunogenic peptides (GIP) to validate GFD adherence. The current study’s adherence proxies did not predict HS. In my opinion, the authors should add objective markers, or at least report them if available, in order to clarify whether “true” adherence can modify HS risk. 

7. Ultrasound (Hamaguchi score) is pragmatic but less sensitive and non-quantitative compared with CAP/FibroScan or MRI-PDFF used in contemporary MASLD research. The authors should discuss any potential expected misclassification, or they could include a subset validation with CAP/MRI-PDFF to benchmark accuracy. 

8. Cross-sectional design precludes causal inference. Furthermore, the cohort is predominantly female and from a single-country. Thus, the authors should discuss these constraints and suggest a longitudinal follow-up focusing on incident MASLD under a nutritionally optimised GFD. 

9. The authors should discuss gut–liver axis, intestinal permeability, and microbiome shifts under GFD/CD in order to propose testable intermediate phenotypes (e.g., LPS, bile acid profiles, SCFAs) for future studies. 

10. The authors should exclude alternative steatotic liver disease etiologies and document alcohol thresholds to align with MASLD/MetALD framework used in the literature. 

Comments on the Quality of English Language

Although the study is well written overall, the authors could polish their phrasing in the Discussion and summarize better the key predictors including all the necessary adjusted odds ratios with the relevant 95%CIs for both the entire and CD-only cohorts. There were ORs in the manuscript that were not accompanied by 95%CIs so they make no sense.

Author Response

Comments 1: The authors should take into consideration the recent shift in the nomenclature from NAFLD to MASLD/MASH. The authors reported HS prevalence but they did not take into consideration the MASLD criteria (e.g., explicit capture of BP, fasting glucose, TG/HDL, or their treatments), thereby limiting comparability to current guidelines and to prior epidemiology. I would recommend that the authors should consider re-analyzing the prevalence using MASLD definitions and reporting overlap etiologies/exclusions.

Response 1: 

We thank the reviewer for this timely and important comment. We fully acknowledge the recent transition in nomenclature from NAFLD to MASLD/MASH and agree that this new framework provides a more comprehensive approach to metabolic-associated liver disease. Based on our available data, we estimated that most patients with hepatic steatosis (HS) in our study would also meet the MASLD criteria: 27/30 (90%) of newly diagnosed CD patients, 42/54 (78%) of CD patients on a GFD, and 13/15 (87%) of controls with HS.

Although a complete re-analysis based on the MASLD definitions could not be done during this revision period, we acknowledge the importance of this updated classification and plan to include MASLD-based analyses in future publications from the ARCTIC cohort.

Comments 2: Based on the previously published literature, the highest incident NAFLD/MASLD risk is reported within the first year after CD diagnosis, persisting long-term, and implicating weight gain and metabolic syndrome after starting GFD. The authors should stratify HS by the time since CD diagnosis and by GFD duration so that they would directly test these signals and better understand the higher HS proportion in pre-GFD patients. 

Response 2: 

We thank the reviewer for this comment. As suggested, we performed an additional analysis stratifying HS by disease duration. We found that a shorter duration of CD was significantly associated with the presence of HS (p = 0.048). Specifically, the median disease duration was 8 years (IQR: 3; 14) in patients without HS and 5.5 years (IQR: 2; 9.75) in those with HS. Notably, in our cohort, the highest prevalence of HS was observed among newly diagnosed CD patients. This finding suggests that HS may occur more frequently during the earlier stages of the disease.

Comments 3: In previously published studies, multiple MASLD pathways were reported that pertained to CD—dietary excess sugar/fat, low fibre, insulin resistance, PPI exposure, physical inactivity, alcohol, and dysbiosis. The authors acknowledged missing key covariates (e.g., physical activity, comorbidities) and found that diabetes was significant in one model; however, should also assess HOMA-IR, PPI use, alcohol intake, or socioeconomic factors, and consider performing a sensitivity analysis including them.

Response 3: 

We thank the reviewer for this important comment. The present work reports partial results from our larger, ongoing multicentre ARCTIC study (NCT05530070). The factors mentioned by the reviewer, including insulin resistance (HOMA-IR), physical activity, and socioeconomic status, are indeed part of the full ARCTIC study protocol and are currently being collected. Therefore, data on these variables will be available upon completion of the full cohort and will be analyzed in future publications. The aim of this nested cross-sectional analysis was not to explore all possible causal pathways but to focus primarily on dietary factors associated with hepatic steatosis in celiac disease.

Comments 4: We already know that GFDs often have higher sugar/fat and lower fibre/micronutrients, with processed gluten-free products as a distinct exposure. A brief MDS and a general “standardized dietitian evaluation” may miss these gradients. I recommend that the authors: a. quantify total energy, macronutrients (especially free sugars, saturated fat), fibre, and the proportion of ultra-processed gluten-free foods; b. report oats/pseudocereals use separately; and c. consider an a-priori “healthy GFD” score aligned with the current recommendations.

We thank the reviewer for these valuable and constructive suggestions. We fully agree that a detailed assessment of energy, macronutrients, fiber, and the consumption of processed gluten-free products provides important nutritional insights. However, in this study, gluten-free diet adherence was evaluated using the Standardized Dietitian Evaluation (SDE), a validated and widely accepted method that has been applied in several previous studies. We acknowledge that a limitation of the current analysis is the lack of data on nutrient composition and processed food consumption.
Within the framework of the ongoing ARCTIC study, a randomized controlled trial is currently being conducted in which participants receive structured dietary counselling aimed at improving the nutritional quality of their GFD and reducing the intake of unfavourable components. The results of this intervention will be reported in future publications.

Comments 5: The authors' suggestion of non-significant trends (e.g., higher odds with whole grains; favorable pattern with olive/rapeseed oil) seems under-powered. Given prior recommendations encouraging naturally gluten-free whole grains and unsaturated fats for MASLD prevention, I think that the authors should avoid over-stating these exploratory signals and rather present them as hypothesis-generating and adjust for multiple testing.

Response 5: 

We thank the reviewer for this important observation. We agree that these findings should be interpreted with caution. Accordingly, we have revised the text to clearly indicate that these associations are exploratory and hypothesis-generating, rather than confirmatory. We have also rephrased the relevant sentences in the Discussion to avoid overstatement and to better reflect the limited power of these analyses.

Comments 6: Previous studies recommended serial serology (tTG/EMA) and, where possible, gluten immunogenic peptides (GIP) to validate GFD adherence. The current study’s adherence proxies did not predict HS. In my opinion, the authors should add objective markers, or at least report them if available, in order to clarify whether “true” adherence can modify HS risk. 

Response 6: 

Thank you for your suggestion. This nested cross-sectional study is part of the ARCTIC study (NCT05530070). Within the framework of this investigation, dietary adherence of patients with celiac disease (CD) is determined using (1) a dietary interview conducted by an expert dietitian applying the Standardized Dietitian Evaluation (SDE) method, (2) coeliac-specific antibodies (tTG and EMA), and (3) urinary gluten immunogenic peptide (GIP) measurement. Consequently, data on tTG/EMA and GIP are available for all CD patients on a gluten-free diet (GFD).

There is no consensus in the literature on which method is the most reliable for assessing adherence. GIP measurement is an objective and sensitive method but reflects only short-term gluten exposure. Serology is also an objective marker of gluten-induced immune response but cannot detect small amounts of inadvertent gluten intake.

In this study, we decided to use the SDE method in the CD on GFD group to assess dietary adherence. This is an accepted and validated method in clinical research.

Among the 156 CD patients on GFD, only 9 were seropositive (tTG and EMA positive) and 12 were GIP positive. Seropositivity and/or GIP positivity were not exclusion criteria. In the CD patients on GFD with steatosis group, 3/54 (5.5%) were tTG/EMA positive and 51/54 (94.5%) were negative. In the CD patients on GFD without steatosis group, 6/102 (5.8%) were tTG/EMA positive and 96/102 (94.2%) were negative.

Regarding GIP positivity, in the CD patients on GFD with steatosis group, 5/54 (9.2%) were GIP positive and 49/54 (90.8%) were negative. In the CD patients on GFD without steatosis group, 7/102 (6.8%) were GIP positive and 95/102 (93.2%) were negative. Following the reviewer’s comment, we performed an additional analysis and found that tTG, EMA, or GIP positivity was not associated with hepatic steatosis; thus, these variables did not influence the results (p>0.05).

Comments 7: Ultrasound (Hamaguchi score) is pragmatic but less sensitive and non-quantitative compared with CAP/FibroScan or MRI-PDFF used in contemporary MASLD research. The authors should discuss any potential expected misclassification, or they could include a subset validation with CAP/MRI-PDFF to benchmark accuracy. 

Response 7: 

We thank the reviewer for this observation. We have added a sentence at the end of the Limitations section, as well as a paragraph in the Discussion to clarify this issue. We agree that liver biopsy remains the gold standard for diagnosing hepatic steatosis and fibrosis. However, it has important limitations, including invasiveness, sampling error due to the structural heterogeneity of diffuse liver diseases, inter-reader variability, and potential complications. These factors, as well as ethical considerations make it unsuitable for screening or for studies requiring repeated assessments.

In our study, we employed ultrasonographic evaluation using the Hamaguchi scoring system, which offers a non-invasive, safe, and widely available alternative. The Hamaguchi score enhances the sensitivity and specificity for identifying metabolic dysfunction–associated steatotic liver disease (MASLD) (>10% steatosis) to 97% and 100%, respectively, compared with liver biopsy. Moreover, this method reduces operator dependency and inter-observer variation, demonstrating accuracy comparable to liver computed tomography.

Ultrasonography is also widely accepted in previous studies as a valid method for the assessment of hepatic steatosis, and it was the imaging modality predefined in our published study protocol (NCT05530070). While advanced modalities such as MRI-PDFF or FibroScan-CAP can also provide accurate quantification, their high cost and limited availability restrict their use in routine clinical and research settings. Therefore, ultrasonography remains the most feasible, validated, and ethically appropriate method for non-invasive assessment of hepatic steatosis in our study population.

Comments 8: Cross-sectional design precludes causal inference. Furthermore, the cohort is predominantly female and from a single-country. Thus, the authors should discuss these constraints and suggest a longitudinal follow-up focusing on incident MASLD under a nutritionally optimised GFD. 

Response 8: 

We thank the reviewer for these valuable comments. We acknowledge that cross-sectional studies do not provide the highest level of evidence and that causal inference cannot be established from this design. However, we did not consider it necessary to emphasize this as a limitation beyond what is inherent to the study design itself.

Although the study was conducted in multiple centers, all participants were recruited in Hungary; therefore, the findings apply primarily to a Central European, predominantly White population. To clarify this, we have specified in the Abstract that the participants were Hungarian, and this information was already included in the Methods section, where we described recruitment from three Hungarian tertiary centers.

We fully agree that longitudinal follow-up is important. In the randomized controlled arm in our ARCTIC study (NCT05530070) we will get high-quality evidence regarding the effectiveness of a 1-year extended, group-based, structured dietary education (with propagation of nutritionally optimized GFD) versus standard of care on body composition- and CV risk-related parameters (including HS) among adult CD patients. Newly diagnosed CD patients are also being followed prospectively to assess the long-term effects of dietary habits. Finally, we note that the female predominance in our sample is consistent with previous epidemiological data showing a 2–3-fold higher prevalence of celiac disease in women.

Comments 9:  The authors should discuss gut–liver axis, intestinal permeability, and microbiome shifts under GFD/CD in order to propose testable intermediate phenotypes (e.g., LPS, bile acid profiles, SCFAs) for future studies. 

Response 9: 

We appreciate this valuable suggestion. We have already discussed the roles of gut–liver axis alterations, intestinal permeability, and microbiome dysbiosis in the pathophysiology of hepatic steatosis in CD in the Discussion section. Specifically, we described how intestinal dysbiosis, leaky gut, and endotoxin-mediated immune activation via Toll-like receptors may contribute to hepatic inflammation and fat accumulation. However, the primary aim of our paper was not to provide a comprehensive literature review on the underlying mechanisms; therefore, we do not intend to expand this part in more detail.

Comments 10: The authors should exclude alternative steatotic liver disease etiologies and document alcohol thresholds to align with MASLD/MetALD framework used in the literature. 

Response 10: 

We thank the reviewer for this important comment. Alcohol consumption was assessed using the Mediterranean Diet Score (MDS) questionnaire, in which both excessive drinkers and abstainers receive a score of 0. This questionnaire was developed before it became widely accepted that any level of alcohol consumption may be harmful; therefore, the categories defined by the MDS do not allow us to precisely distinguish the role of alcohol intake in hepatic steatosis.

The main objective of our study was not to differentiate non-alcoholic from alcohol-related hepatic steatosis, but rather to document the presence of hepatic steatosis (HS) in celiac disease, regardless of etiology. HS can have multiple causes, even in healthy individuals, including altered intestinal permeability and metabolic changes, which may also be influenced by alcohol intake.

Furthermore, the current MASLD/MetALD nomenclature reflects the multifactorial nature of steatotic liver disease, where alcohol is considered one of several contributing factors rather than a strict exclusion criterion. This evolving framework supports our approach of reporting HS prevalence independent of specific etiological classification.

In line with the reviewer’s suggestion, we have also added a clarifying sentence in the Limitations section, noting that alcohol intake could not be quantified precisely using the MDS and that differentiation between alcohol-related and non-alcohol-related steatosis was therefore not feasible.

Comments 11: Although the study is well written overall, the authors could polish their phrasing in the Discussion and summarize better the key predictors including all the necessary adjusted odds ratios with the relevant 95%CIs for both the entire and CD-only cohorts. There were ORs in the manuscript that were not accompanied by 95%CIs so they make no sense.

Response 11: 

We thank the reviewer for this valuable suggestion. We have revised the Results and Discussion sections to ensure that all odds ratios are accompanied by their corresponding 95% confidence intervals and p-values. We have also refined the phrasing in the Discussion to provide a clearer summary of the key predictors identified in both the overall and CD-only cohorts.

Round 2

Reviewer 3 Report

Comments and Suggestions for Authors

Well done